# Assessment of treatment outcomes and associated factors among patients treated with clozapine at amanuel mental specialized hospital, Addis Ababa, Ethiopia

Zehara Reshid[1], Kibrom Haile[1]*, Getinet Ayano[2], Alem Kebede[1], Aynalem Biru[1], Zegeye Yohannis[1]

1 Research and Training Department, St Amanuel Mental Specialized Hospital, Addis Ababa, Ethiopia,
2 PostDoc Position at Curtin University, Bentley Washington, Australia

* kibromhaile4@gmail.com

## Abstract

### Background

Clozapine is a medication used in psychiatry. It is the first atypical antipsychotic drug and the most effective in treatment resistant cases of schizophrenia and schizoaffective disorder. Studies also show benefits of clozapine in patients with tardive dyskinesia. However, there is scarcity of studies which show the outcome of clozapine treatment and factors associated with it in the Ethiopian context.

### Objective

The aim of the study was to assess treatment outcome and factors associated with it among patients treated with clozapine at Amanuel Mental Specialized Hospital Addis Ababa, Ethiopia.

### Method

A hospital-based retrospective study, compounded by cross-sectional design was conducted from June 1–30, 2022. A total of 71 clozapine treated patients were taken through census method. An interviewer administered structured questionnaire, retrospective data from patient medical records, and clinical evaluation of severity and improvement (CGI) were used to collect data. The collected data were entered into Epidata version 4.6 Software then exported to the Statistical Package for the Social Sciences (SPSS) version 25 for statistical analysis. Bivariable and multivariable logistic regression analyses were used to identify factors associated with treatment outcomes. P-values less than 0.05 were considered statistically significant and strength of the associations were presented by adjusted odds ratios with 95% confidence intervals.

**Data availability statement:** The data underlying this study cannot be shared publicly as they contain potentially identifying or sensitive patient information, making it challenging to share them without violating ethical principles. However, the data will be made available upon reasonable request from the corresponding author. Requests for data access can be directed to the ethical review committee at the following institutional address: amsh@res.gov.et.

**Funding:** The author(s) received no specific funding for this work.

**Competing interests:** The authors have declared that no competing interests exist.

**Abbreviations:** AIMS, Abnormal Involuntary Movement Scale; AMSH, Amanuel Mental Specialized Hospital; AOR, Adjusted Odds Ratio; CGI, Clinical Global Impression; CI, Confidence Interval; COR, Crude Odds Ratio; OSS-3, Oslo Social Support Scale; PANSS, Positive and Negative Symptom Scale; SPSS, Statistical Package for Social Sciences; TD, Tardive Dyskinesia; TRS, Treatment-Resistant Schizophrenia.

## Results

Fifty one (71.8%) of participants showed improvement by clozapine treatment. Literacy level of high school and above showed significant positive association with treatment outcome (AOR: 7.65, 95% CI: 1.26–46.36, p<0.05) while extreme severity of illness at onset of clozapine treatment showed significant negative association with treatment outcomes (AOR: 0.13, 95% CI: 0.02–0.99, p<0.05).

## Conclusions

Treatment with clozapine resulted in significant improvement in clinical and functional outcomes of treatment-resistant schizophrenia and schizoaffective disorder, as well as in tardive dyskinesia. Higher literacy level and severity of illness at onset of treatment showed significant associations with outcome.

Clozapine is an atypical or second generation antipsychotic first used in the 1960s. It was withdrawn at first after its use was associated with a number of deaths due to agranulocytosis in Finland in 1975 [1,2]. In 1988 a landmark study demonstrated that the medicine was helpful to patients with schizophrenia who were unresponsive to other medications. Clozapine was then reintroduced into clinical use [3]. Since then, clozapine has been shown to be the only medicine which reduced suicidal behavior in patients with schizophrenia. Clozapine is now the medicine of choice for treatment-resistant schizophrenia [1]. The antipsychotic efficacy and the complete absence of extrapyramidal adverse effects of clozapine have been well documented. The benefit of clozapine for treating patients with treatment-resistant schizophrenia and those who are particularly prone to extrapyramidal adverse effects has been most remarkable [2].

## Background

Schizophrenia is one of the most severe mental disorders. It has a life-time prevalence of 4.8–7.2 per 1000. Schizophrenia causes huge disease burden and it is associated with a high number of disability-adjusted life years. Families of patients with schizophrenia experience the burden of the illness in many ways; this includes grief associated with knowing about the diagnosis, direct costs of treatment, lost income in taking care of the patient, stigma, and the feeling of guilt about their role in the etiology of the illness. The burden is particularly great in relatives of patients with treatment-resistant illness [4].

Treatment-resistant schizophrenia is defined by lack of response to treatment with at least two antipsychotics which were given at appropriate doses and administered for at least 6 weeks each [5]. The international guidelines by the Treatment Response and Resistance in Psychosis (TRRIP) Working Group provide a clear consensus definition of treatment-resistant schizophrenia (TRS) [6,7]. According to the guidelines the diagnosis of TRS requires fulfillment of the following conditions: there should be persistent positive, negative and cognitive symptoms of schizophrenia for at least three months of duration having at least moderate severity and functional impairment; there should be insufficient response to treatment with at least two different antipsychotic drugs at adequate doses with at least six weeks duration of treatment each (adequate doses refer to dose equivalents of at least chlorpromazine 600mg per day); finally, there should be confirmation of sufficient treatment adherence to treatment by the patient as defined by patients having taken at least 80% of the prescribed doses [6,7].

About two-thirds of all patients who have the diagnosis of schizophrenia suffer from a recurrent or chronic course of the illness, and about 30% of all patients with schizophrenia develop TRS [6,8]. Currently, clozapine remains the only effective antipsychotic drug for patients with TRS [6,9–11]. Unequivocal evidence supports the superior efficacy of clozapine for the reduction of positive symptoms and global psychopathology in TRS compared with other antipsychotics [6,8]. Treatment with clozapine resulted in improved treatment adherence and reduced rehospitalization rates among patients [6,8].

Clozapine improved social and occupational functioning and quality of life; it also reduced extrapyramidal symptoms including tardive dyskinesia (TD) in patients [8]. Tardive dyskinesia (TD) is an antipsychotic-induced movement disorder that typically occurs after long-term exposure to antipsychotic drugs [12]. This disorder is characterized by abnormal involuntary movements of the facial, perioral, neck, back, and extremities muscles. It is a distressing and disabling disorder. Studies have shown that clozapine showed beneficial effects in patients with TD [13,14].

The treatment outcomes of patients treated with clozapine is not well known in Ethiopia. There is scarcity of studies conducted among patients treated with clozapine in the Ethiopian context. Therefore this study was conducted to fill this gap by generating evidence about outcome of treatment with clozapine at Amanuel Mental Specialized Hospital.

## Materials and methods

### Study area and period

The research was carried out at Amanuel Mental Specialized Hospital (AMSH) from June 1–30, 2022. Amanuel Mental Specialized Hospital is located in Addis Ababa, the largest city and the capital of Ethiopia. As of the latest 2021 population census conducted by Ethiopian National Statistics Authorities, the metro area population of Addis Ababa was 5,006,000. However, AMSH was the final referral center for patients with mental illness not only from Addis Ababa, but from all areas of Ethiopia. The hospital had 239 beds for admitting patients for inpatient treatment. The average length of stay for patients with mental illness in the hospital's inpatient wards was 32–35 days. The hospital also provided outpatient and emergency services mainly for people with mental illness. The average daily visit to the outpatient unit was around 500, including visits to the emergency department which averages 20 per day. Clozapine treatment was introduced to the hospital service in 2016.

When treatment with clozapine was introduced as a formal treatment in the hospital, a treatment guideline was prepared and a team of professionals with full time dedication were assigned for its safe administration and the disciplined implementation of the guideline. Clear criteria for recruiting patients for clozapine treatment were part of the guideline. The definition of treatment resistance was strictly followed with at least one of the failed antipsychotic medications being a second-generation antipsychotic such as risperidone and olanzapine. The treating team was made to be led by a dedicated expert psychiatrist who was authorized as the only person who could prescribe the drug. The team included a clinical pharmacist who also was authorized as the sole dispenser of clozapine. Competent nurses were chosen and were included in the team. Admission to hospital was mandatory for initiation of clozapine; once clozapine treatment was initiated the patients stayed in hospital for minimum of one month before discharged. Admission allowed for safe initiation and proper escalation of clozapine. It also allowed for close monitoring of patients for side effects and for the weekly blood tests of complete blood count (CBC) and erythrocyte sedimentation rate (ESR). The hospital did not have access to measurement of serum levels of clozapine for monitoring therapeutic levels. Therefore, the escalation of clozapine dose was made solely by clinical evaluation of progress

and/or onset of adverse effects. To ensure good adherence during clozapine treatment, patients with a trusted caregiver were initiated on the medication. The caregivers were given health education about the treatment, possible adverse effects and the importance of adherence. The caregivers supported the patients to adhere to treatment after they were convinced that clozapine was the only remaining hope for better outcomes of the patients.

## Study design and study population

A hospital-based retrospective study design, compounded by cross-sectional study design was used. The socio-demographic data, the level of social support, the level of daily stressful life events and the use of substances were gathered by interviewing participants cross-sectionally at the time of data collection. The clinical data were gathered retrospectively from medical records of the participants. All patients who had treatment with and follow-up for clozapine at Amanuel Mental Specialized Hospital were the source population for this study. All patients who had follow-up for clozapine treatment for at least three months at Amanuel Mental Specialized Hospital were recruited for the study.

## Sample size determination

All patients on clozapine treatment who were eligible for the study were included for the study. Due to limited number of patients who were on clozapine treatment, it was decided to include for the study all those who had had follow-up for at least 3 months.

## Data collection and data collection instruments

A semi-structured questionnaire was prepared to collect socio-demographic data, clinical factors, and substance use factors of participants both from medical records and by interviewing participants. Medical records of participants were used to get retrospective clinical data. Expert psychiatrist scored the CGI severity and improvement scale of each participant on medical charts during the data extraction period, which lasted from June 1–30, 2022. Data was collected after the CGI scores were scored for each participant. Variable selection was based on existing literature as well as clinical experience; the decision to include the selected variables was made after discussion and agreement among the authors. Data collectors accessed medical records of participants and extracted relevant retrospective clinical data. Then data were filled on to coded data filling formats which had no patient identifying information on them. Authors had no access to information that could identify individual participants. Oslo Social Support Scale (OSS-3) was used to assess for the level of social support of participants. The OSS-3 assessment was conducted by interviewing participants at the time of data collection. OSS-3 consists of 3 items and can have a score range of 3–14. A score of 3–8 indicates poor support, 9–11 shows moderate support and score of 12–14 shows strong support [15]. (Abiola, 2013 #20;Abiola, 2013 #1311)) The daily stressful life events measurement scale was used to identify stressful life events of the study participants. The daily stressful life events data was collected by interviewing participants during the data collection.

The Clinical Global Impression (CGI) instrument was used to assess illness severity at onset of treatment with clozapine and its improvement after treatment with clozapine. The CGI global improvement rate was used as outcome variable. The CGI severity and improvement scales offer a readily understood, practical measurement tool that can easily be administered by a clinician in a busy clinical practice setting [16]. It has CGI-Severity (CGI-S) scale, which is scored on a 7-point scale; the CGI-Improvement (CGI-I) scale, which also is scored on a 7-point scale from 1= very much improved since initiation of treatment, through 7= very much worse since initiation of treatment [16]. The CGI global improvement rate was made

to have two outcomes by merging "much improved" and "very much improved" together and the "minimally improved" and all the "worsened" together. Those which were rated "much improved" and "very much improved" comprise the "improved" category after merging and the rest comprise the "not improved" category.

Data was collected by interviewing patients by trained psychiatry professionals using semi-structured questioners, by checking patient medical records, and by conducting clinical evaluation of participants by an experienced expert psychiatrist.

## Data quality assurance

In order to maintain data quality, the questionnaire was translated to the local language (Amharic) in order to be understood by all study participants and translated back to English to check for consistency. The Amharic version of the questionnaire was used for interviewing participants. The principal investigator of the study gave a half day orientation to the data collectors and supervisors about the data collection tool, the purpose of the study, the data collection procedure and how to solve problems that occurred during data collection. Before the actual data collection the questionnaires was checked for clarity and comprehensiveness. Pilot study was done on five participants two days before the start of actual data collection; the results were not included in the main study. The data collectors were supervised daily and the filled questionnaires were checked for completeness by the supervisors and the principal investigator.

## Data processing and analysis

Data were coded and entered into Epi-data Version 4.6 Software; then were exported to SPSS version 25 for statistical analysis. Before the analysis, data were checked for completeness and consistency. Descriptive statistics was used to explain the study participants about study variables. Bi-variate and multivariate logistic regression analysis was conducted to identify factors associated with clinical improvement after clozapine treatment. Logistic regression analysis results were reported in crude and adjusted odds ratios and 95% confidence intervals. A P-value of less than 0.05 was considered as statistically significant.

## Ethical consideration

Ethical clearance was obtained from the Institutional Review Board (IRB) of Amanuel Mental Specialized Hospital. Informed consent was obtained from each study participant just before data collection. The participants gave verbal consent to participate and for their clinical records to be used in the study. The study participants were informed that participating in the research was fully voluntary and they could withdraw from the study at any time during the data collection. Participants of the study were also informed that their identity will not be revealed in anyway. They were also informed that there would be no monetary or other direct benefits they would get by participating in the study. To keep the confidentiality of the study participants' personal identifiers were not included in the data collection format.

## Results and discussion

### Results

**Socio-demographic characteristics of the study participants.** A total of 71 participants were included in the study. The mean (±SD) age of participants was 36.4 (11.1) years. Among the study participants majority, 42 (59.2%) were in the age group 30–49 years, 54 (76.1%) were male in sex, 35 (49.3%) were Orthodox Christian in religion and 54 (76.1%) were single.

Majority of the study participants had education up to secondary level, 37 (52.1%). Majority of the study participants, 56 (78.9%) were unemployed. Around 65 (91.5%) of the study participants lived with their family (Table 1).

**Clinical characteristics.**  Among the study participants 29 (40.8%) clients were diagnosed with mental illness before 10 years. Around 35 (49%) of the clients started clozapine because of TD and 67 (94.4%) of the clients were diagnosed with schizophrenia. All the patients with TD had the psychiatric diagnosis of schizophrenia. In 29 (40.8%) of participants, the duration of clozapine treatment was one year and above, and in 22 (31%) the dose of clozapine at the time of the study was above 400 mg per day. The mean dose of clozapine was 296 mg. There was no adherence problem reported about monitoring of blood counts. Majority, 49 (69%) of study participants reported no clozapine side effects. No agranulocytosis was identified among study participants; milder side effects such as hypersalivation, sedation and occasional urinary incontinence were reported (Table 2).

**Substance-related characteristics.**  Among the study participants 17 (24%) ever consumed psychoactive substances and 60 (85%) of the participants did not consume psychoactive substances after initiation of clozapine (Table 3).

**Level of psychosocial support and stressful life events.**  Among the participants 33 (46.5%) had moderate social support and 40 (56.3%) of the participants reported they did not face any stressful life events in the past 6 months (Table 4).

**Clozapine treatment outcome.**  Among the total participants 51 (71.8%) showed improvement after they were treated with clozapine. Some 20 (28.2%) participants did not improve with clozapine treatment. In those with treatment resistant schizophrenia and schizoaffective disorder, 28 of the 36 participants (77.8%) were improved with clozapine

**Table 1.  Socio-demographic characteristics of participants who were treated with clozapine in Amanuel Mental Specialized Hospital, Ethiopia.**

| Variable | Categories | Frequency | Percentage (%) |
|---|---|---|---|
| Sex | Male | 54 | 76% |
| | Female | 17 | 24% |
| Age | 18-28 years | 18 | 25.4% |
| | 29-38 years | 29 | 40.8% |
| | 39-48 years | 13 | 18.3% |
| | 49 years and above | 11 | 15.5% |
| Religion | Orthodox | 35 | 49.3% |
| | Christian | 21 | 29.6% |
| | Muslim | 14 | 19.7% |
| | Protestant | 1 | 1.4% |
| | Others | | |
| Marital status | Single | 54 | 76.1% |
| | Married | 7 | 9.9% |
| | Divorces | 8 | 11.3% |
| | Windowed | 2 | 2.8% |
| Education level | Illiterate | 4 | 5.6% |
| | 1-8 grade | 11 | 15.5% |
| | 9-12 grade | 37 | 52.1% |
| | Diploma and above | 19 | 26.8% |
| Occupation | Governmental | 6 | 8.5% |
| | Private employee | 9 | 1.7% |
| | Jobless | 56 | 78.9% |
| Living condition | Live with family | 65 | 91.5% |
| | Living alone | 5 | 7% |
| | Live with relative | 1 | 1.4% |

**Table 2. Clinical characteristics of participants who were treated with clozapine in Amanuel Mental Specialized Hospital, Ethiopia.**

| Variable | Categories | Frequency | Percentage (%) |
|---|---|---|---|
| Major diagnosis | Schizophrenia | 67 | 94.4% |
| | Schizoaffective disorder | 4 | 5.6% |
| Duration since initial diagnosis of the illness | 1 year and below | 7 | 9.9% |
| | Above 1–5 years | 25 | 35.2% |
| | Above 5–10 years | 10 | 14.1% |
| | above 10 years | 29 | 40.8% |
| Reason for starting Clozapine | TRS (schizophrenia) | 32 | 45.1% |
| | TRS(schizoaffective disorder) | 4 | 5.6% |
| | Drug side effect(TD) | 35 | 49.3% |
| Duration of treatment with Clozapine | 6 months and below | 26 | 36.6% |
| | 7-12 months | 16 | 22.5% |
| | Above 12 months | 29 | 40.8% |
| Current dose of Clozapine | 100 -150 mg | 8 | 11.3% |
| | 200- 250 mg | 22 | 31% |
| | 300- 350 mg | 19 | 26.8% |
| | 400 mg and above | 22 | 31% |
| Reported Clozapine side effect | Yes | 22 | 31% |
| | No | 49 | 69% |

**Table 3. Substance-related characteristics of participants who were treated with clozapine in Amanuel Mental Specialized Hospital, Ethiopia.**

| Variable | Categories | Frequency | Percentage (%) |
|---|---|---|---|
| Ever used psychoactive substance | Yes | 17 | 23.9% |
| | No | 54 | 76.1% |
| Used substance after initiation of clozapine | Yes | 11 | 15.5% |
| | No | 60 | 84.5% |
| Type of substance Currently consumed | No | 60 | 84.5% |
| | Cigarette | 6 | 8.5% |
| | Khat | 3 | 4.2% |
| | Both Khat and cigarette | 2 | 2.8% |

**Table 4. Social support and stressful events among participants who were treated with clozapine in Amanuel Mental Specialized Hospital, Ethiopia.**

| Variable | Categories | Frequency | Percentage (%) |
|---|---|---|---|
| Social support | Poor social support | 23 | 32.4% |
| | Moderate social | 33 | 46.5% |
| | Strong social support | 15 | 21.1% |
| Stressful life events | Stressful life event | 31 | 43.7% |
| | No stressful life event | 40 | 56.3% |

treatment. In participants who were diagnosed with TD the response rate was 23 out of 35 (65.7%).

**Factors associated with outcome of treatment with clozapine.** On multiple logistic regression analysis there was a significant improvement to clozapine treatment based on CGI scores among participants with literacy level of high school and above (adjusted odds ratio (AOR): 7.65, 95% CI: 1.26–46.36, p<0.05). However, there was significantly low improvement to clozapine treatment among those who had very severe illness at onset of treatment compared to those who had mild to marked illness (AOR: 0.13, 95% CI: 0.02–0.99, p<0.05) (Table 5). Despite not significant on multiple logistic regression analysis, there was a tendency for increased improvement with clozapine treatment among those who were treated for more

**Table 5. Crude and adjusted odds ratios of factors associated with the CGI scores of improvement with clozapine treatment among study participants who were treated with clozapine in Amanuel Mental Specialized Hospital, Ethiopia.**

| Characteristics | Categories | Improved | Not improved | COR (95%CI) | AOR(95%CI) |
|---|---|---|---|---|---|
| Age | Up to 40 years | 32 | 16 | 1 | 1 |
| | Above 40 years | 19 | 4 | 2.38 (0.69, 8.16) | 2.49 (0.59, 10.63) |
| Sex | Male | 42 | 12 | 1 | 1 |
| | Female | 9 | 8 | 0.32 (0.10, 1.01) | 0.35 (0.09, 1.39) |
| Religion | Muslim | 18 | 3 | 1 | 1 |
| | Christian | 33 | 17 | 0.32 (0.08, 1.25) | 0.25 (0.04, 1.82) |
| Duration of clozapine treatment | Up to 12 months | 26 | 16 | 1 | 1 |
| | More than 12 months | 25 | 4 | 3.85 (1.13, 13.10)* | 1.64 (0.35, 7.7) |
| Current dose | Up to 300 mg | 31 | 16 | 1 | 1 |
| | Above 300 mg | 20 | 4 | 2.58 (0.75, 8.84) | 4.27 (0.72, 25.45) |
| Severity of illness | Mild to marked | 32 | 10 | 1 | 1 |
| | Severe | 13 | 5 | 0.81 (0.23, 2.84) | 0.81 (0.19, 3.48) |
| | Extremely severe | 6 | 5 | 0.38 (0.09, 1.50) | 0.13 (0.02, 0.99)* |
| Literacy level | Elementary and below | 7 | 8 | 1 | 1 |
| | Highschool and above | 44 | 12 | 4.19 (1.26, 13.89)* | 7.65 (1.26, 46.36)* |

*P< 0.05 - Statistically significant association

than 12 months (crude odds ratio (COR): 3.85, 95% CI: 1.13–13.10, p<0.05). Age, sex, religion, current dose of clozapine, did not show significant association with level of improvement to clozapine treatment based on CGI scores (Table 5).

## Discussion

The aim of this study was to assess clozapine treatment outcomes of patients at Amanuel mental specialized hospital. In this study the indications for treatment with clozapine were treatment resistant schizophrenia, treatment resistant schizoaffective disorder and tardive dyskinesia (TD).

In our study, majority (71.8%) of clozapine-treated patients had improvement on the GCI improvement rating, with slightly higher improvement rating in its use in treatment-resistant schizophrenia and schizoaffective disorder (77.8%) than in TD (65.7%). The patients were previously treated with other medications without response, and this means that clozapine was the only option they had to show the level of improvement in their clinical symptoms and level of function. This is indicative of how much clozapie is important in the management of such patients in any treatment facility. Our findings are better than a previous study conducted in Turkey among TRS patients treated with clozapine who showed clinical response in 55.7% [17]. Our finding about the effectiveness of clozapine is TD is also supported by previous studies [12,18,19].

Our findings showed significant positive association of higher education level with response to clozapine treatment compared with lower education level, with those having high school and above significantly associated with improvement on the CGI scale. This could be due to better understanding and acceptance of the treatment in this population group which could have resulted in better compliance. Another factor which showed significant association with improvement on CGI scores is severity of illness at onset of treatment. Those who had extremely severe illness at onset of treatment with clozapine were more likely to show less improvement on CGI scores compared to those who had mild to marked severity. This finding suggests that

treatment is better initiated before the illness severity becomes extreme. However, duration of illness at onset of clozapine treatment did not show any association with CGI improvement scores and was not included in the multivariate binary logistic regression analysis.

Even if it did not show significant association in the adjusted odds ratios, duration of treatment had a tendency to be associated positively with CGI improvement rating on the crude odds ratio. In line with this, a study conducted in New Zealand showed that increasing treatment duration led to higher improvements in clinical and functional outcomes [20]. This suggests that prolonged clozapine trials are needed to improve outcomes in the most seriously ill patients with diagnoses of treatment-resistant schizophrenia, schizoaffective disorder and tardive dyskinesia.

There was one previous study conducted in Ethiopia and it showed remarkable decrease in severity scales of CGI and AIMS among patients with TRS and TD treated with clozapine in a general hospital [21]. The findings of the study agree with our findings. Our study, however, included bigger sample size than this previous study. Our study is also different from this previous study in that it included factors associated with outcomes of the CGI.

**Strengths and limitations of the study.**

**Strengths:** The study is one of few in Ethiopia, and tried to measure the frequency of response among those treated with clozapine. The study also tried to identify factors associated with chances of improvement in outcomes of clozapine treatment.

**Limitations:** The sample size was small and impacted the strength of statistical measures. Records of positive and negative symptom scale (PANSS) and abnormal involuntary movement scale (AIMS) were not found for many patients making it difficult to measure differences after onset of clozapine treatment in the parameters. This made us rely on CGI scores as improvement measures. The retrospective study design had imposed its limitations since the PANSS and AIMS measures were not consistently used to record accurate scores for purposes of research. The other limitation was the absence of measurement of therapeutic clozapine levels. This made monitoring of adherence challenging during treatment.

## Conclusions

The study showed that treatment with clozapine resulted in significant improvement in clinical and functional outcomes in participants who were diagnosed with treatment resistant schizophrenia and treatment resistant schizoaffective disorder, as well as in those who had tardive dyskinesia. Higher literacy level predicted better chances of improvement, while extremely severe illness at onset of clozapine treatment predicted lesser chances of improvement. Further studies with bigger sample size are recommended. Randomized controlled trials are recommended to verify the findings of this study in the Ethiopian and African context.

## Acknowledgments

First, we would like to express our deepest gratitude to Amanuel Mental Specialized Hospital for allowing and facilitating the study. Then, we would like to thank all our colleagues who supported us during the time of conducting this research. We also thank the data collectors. Our best regards, however, go to the study participants for devoting their time and energy providing all the necessary information.

## Author contributions

**Conceptualization:** Zehara Reshid, Kibrom Haile, Alem Kebede, Aynalem Biru, Zegeye Yohannis.

**Data curation:** Zehara Reshid, Kibrom Haile.

**Formal analysis:** Zehara Reshid, Kibrom Haile, Getinet Ayano, Alem Kebede, Aynalem Biru, Zegeye Yohannis.

**Investigation:** Zehara Reshid, Alem Kebede, Aynalem Biru, Zegeye Yohannis.

**Methodology:** Zehara Reshid, Kibrom Haile, Getinet Ayano, Alem Kebede, Aynalem Biru, Zegeye Yohannis.

**Project administration:** Zehara Reshid.

**Resources:** Zehara Reshid.

**Supervision:** Zehara Reshid, Alem Kebede, Aynalem Biru, Zegeye Yohannis.

**Validation:** Kibrom Haile, Getinet Ayano, Zegeye Yohannis.

**Visualization:** Zehara Reshid.

**Writing – original draft:** Zehara Reshid, Kibrom Haile, Getinet Ayano, Alem Kebede, Aynalem Biru, Zegeye Yohannis.

**Writing – review & editing:** Kibrom Haile.

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
