## [Decision Letter · Decision Letter 0]

9 Sep 2024

PONE-D-24-21600Assessment of Treatment Outcomes and Associated Factors among Patients Treated with Clozapine at Amanuel Mental Specialized Hospital, Addis Ababa, Ethiopia.PLOS ONE

Dear Dr. Haile,

Thank you for submitting your manuscript to PLOS ONE. After careful consideration, we feel that it has merit but does not fully meet PLOS ONE’s publication criteria as it currently stands. Therefore, we invite you to submit a revised version of the manuscript that addresses the points raised during the review process.

 Please submit your revised manuscript by Oct 24 2024 11:59PM. If you will need more time than this to complete your revisions, please reply to this message or contact the journal office at plosone@plos.org . Please include the following items when submitting your revised manuscript:

We look forward to receiving your revised manuscript.

Kind regards,

Rishab Gupta, MD

Academic Editor

PLOS ONE

Journal Requirements:

2.  In the ethics statement in the Methods, you have specified that verbal consent was obtained. Please provide additional details regarding how this consent was documented and witnessed, and state whether this was approved by the IRB. 

"First, we would like to express our deepest gratitude to Amanuel Mental Specialized Hospital for financial support. Then, we would like to thank all our colleagues who supported us during the time of conducting this research. We also thank the data collectors. Our best regards, however, go to the study participants for devoting their time and energy providing all the necessary information. "

4. We note that your Data Availability Statement is currently as follows: 

"All relevant data are within the manuscript and its Supporting Information files."

**Additional Editor Comments:**

Dear Authors,

I have gone through your paper, and the reviewers' comments. Based on that, I recommend that you revise the paper and address reviewers' concerns, and also address the language issues with the manuscript.

Reviewers' comments:

Reviewer's Responses to Questions

**Comments to the Author**

1. Is the manuscript technically sound, and do the data support the conclusions?

Reviewer #1: Yes

Reviewer #2: Yes

2. Has the statistical analysis been performed appropriately and rigorously? 

Reviewer #1: Yes

Reviewer #2: Yes

3. Have the authors made all data underlying the findings in their manuscript fully available?

Reviewer #1: Yes

Reviewer #2: Yes

4. Is the manuscript presented in an intelligible fashion and written in standard English?

Reviewer #1: No

Reviewer #2: Yes

5. Review Comments to the Author

Reviewer #1: It is commendable that the authors have attempted to address this topic within the Ethiopian context. However, given the current state of the manuscript, it may require further development to make the manuscript robust.

While the treatment outcome and factors associated with improvement in the outcome of schizophrenia have been well studied, this study attempts to emphasize these aspects in an Ethiopian context. It is interesting to note that clozapine has only been used since 2016 in the hospital. While the study is important from a contextual relevance and representation point of view, I share my comments below about major concerns in the methodology and scientific rigor used in the study.

Methodology

1. Sample Size and Duration

• While all eligible patients were included in the study, there is no clear rationale provided for the chosen sample size and the one-month duration of data collection. Implementing an a priori sample size calculation would enhance the study’s rigor and validity.

2. Primary Outcome Measures:

• The reliance on Clinical Global Impression (CGI) scores as the primary outcome measure, without the consistent use of more specific tools such as the Positive and Negative Syndrome Scale (PANSS) or the Abnormal Involuntary Movement Scale (AIMS), limits the ability to fully capture the range of patient improvements and side effects. Incorporating a wider range of validated psychiatric scales could provide a more nuanced understanding of treatment outcomes.

3. Variable Selection

• The variables chosen for the study seem arbitrary, and the reasons for selecting these particular factors have not been justified by the authors. Providing context-specific reasons relevant to Ethiopia would further strengthen the study’s rationale, especially given the recent introduction of clozapine in the hospital since 2016.

Results

1. Variable Representation:

• The representation of many variables in the analysis and results sections is loose, which reduces the study's focus. For example, the data mentioned in Table 2 does not adequately describe the population or bring clinical relevance to understanding the study. Similarly, the effect of substance use, as described in Table 3, is another variable that describes the population but has not been explored further in the study.

Discussion

1. Contextualization of Findings:

• The discussion section does not sufficiently contextualize the findings within the broader body of literature on clozapine's use in similar populations. Comparing the results with other studies conducted in different settings or populations would help in understanding the uniqueness or commonality of the findings. A more thorough review of existing literature is needed to place the study’s results in a broader clinical and research context.

Conclusion

1. Limitations and Practical Implications:

• The conclusion makes sweeping claims without adequately addressing the study’s limitations, such as the small sample size and retrospective design. It fails to offer practical implications for clinical practice or specific directions for future research. Additionally, the conclusion does not adequately address the long-term implications of clozapine treatment, which diminishes its relevance in clinical settings.

Reviewer #2: The study addresses an important area that has been understudied in LMIC. I have the following concerns.

The methods are not sufficiently described to understand whether this was a retrospective or cross sectional study. It is not clear as to which of the data points were collected through interviews and which of them obtained retrospectively. The authors should specify this clearly in the manuscript. The abstract mentions the data was collected retrospectively.

What was the mean duration of hospitalization for the patients?

How was adherence ensured post-discharge?

Was there access to Therapeutic clozapine level measurements?

What was the psychiatric diagnostic breakup of the TD patients?

What was the pathway to clozapine (drugs used before clozapine was initiated)

6. PLOS authors have the option to publish the peer review history of their article (what does this mean? ). If published, this will include your full peer review and any attached files.

**Do you want your identity to be public for this peer review?** For information about this choice, including consent withdrawal, please see our Privacy Policy .

Reviewer #1: No

Reviewer #2: **Yes: ** Krishna Prasad Muliyala

---

## [Author Response · Author response to Decision Letter 1]

13 Sep 2024

Response to reviewers

We want to thank the reviewers for critically examining the manuscript and providing valuable comments. We are confident that the comments will greatly improve the quality of the manuscript. We have addressed the comments and improved the manuscript accordingly to the best of our capacity.

Our responses:

Reviewer 1:

Methods:

1.Sample size and duration: Due to limited number of patients who were on treatment with clozapine in the hospital, we decided to include all those eligible in the study. We are aware that this will be a limitation of the study and have acknowledged this fact in the discussion. The duration for the study was the time taken to include all eligible participants when they visited the hospital for follow-up. We have added statements in the manuscript addressing this issue for the purpose of clarity.

2.Using CGI without consistent use of PANSS and AIMS: Since this is a retrospective study compounded by cross-sectional design, the use of assessment tools such as PANSS and AIMS were not used consistently during treatment. However, thorough clinical assessments were made at the time of all progress and follow-up evaluations. Again, this is a limitation of the study and we acknowledged this in the limitations section as you suggested. We suggest experimental studies with controls to verify the findings of our study and included such suggestion in the conclusion part.

3.Variable selection: The variable selection was not arbitrary but based on factors described in the literature; additional factors which were considered to impact treatment outcome of clozapine from clinical experience were included. We did not find several studies which described factors associated with outcome of clozapine treatment in the literature.

Results:

1.Variable representation: the variables included in the study are all relevant to the study because the study explores not only outcome of clozapine treatment but also factors associated with good outcome. Several variables were included in order to identify the factors which had true effect on outcome of treatment from a relatively long list of factors. Therefore, we don’t believe including those variables would affect focus of the study. The clinical characteristics of participants listed in Table 2 included factors which were considered to affect treatment outcomes. Along with socio-demographic, stress and social support, and substance use characteristics, these clinical characteristics can describe the study participants. The table, however, shows the clinical characteristics most relevant to the study. The effect of substance use was analyzed to check for presence of associations with treatment outcomes. However, the study did not find any associations of substance use. Therefore, substance use was included for exploration and was explored but did not show any association. That was why it was not discussed further in the results and discussion.

Discussion:

1.Contextualization of findings: As you commented, we have found and included a previous study conducted in Ethiopia to contextualize our findings. We believe the inclusion of this study in the discussion section will greatly address the issue of contextualizing the findings of our study. We have also discussed how our study differs from this previous study. We have included this in the revised manuscript.

Conclusion:

1.Limitations and practical implications: We have addressed the small sample size in the limitations section, as well as elaborated it in the methods section. We have added the limitations due to the study design (retrospective) in the limitations section based on your comment. We believe readers would understand the limitaions from the elaborations in the methods section and we also emphasized them in the limitations section. But we believe our study doesn’t fail to offer practical implications for clinical practice as the findings of the study clearly show the benefits of clozapine. We have also added the importance of randomized clinical trials to verify the findings of our study given its limitations. We have recommended future well designed studies. Our study is just one study and further studies will provide further evidence for clinical and academic purposes. As for addressing the long-term implication of clozapine treatment, we leave it for future studies because it is not the objective of our study to show the long-term effects of treatment with clozapine.

Reviewer 2:

Based on your comments we have elaborated the methods section. We have addressed your comments as follows:

We have modified the study design and included it in our revised manuscript as “… retrospective study design compounded by cross-sectional design” We have also specified which data were retrospective and which ones were cross-sectional in the relevant areas of the methods section.

We have made amendments to the abstract section as well.

The mean duration of hospitalization was between 32 and 35, but the guideline recommended minimum of one month. Patients were discharged after one month except they can take few days during discharge process.

We have addressed how adherence was ensured after discharge by stating “… based on the guideline… patients with a trusted caregiver were initiated on clozapine… caregivers were given health education… caregivers supported patients to adhere to treatment after they were convinced clozapine was the only remaining hope for better outcome.”

There was no access to measures of therapeutic levels of clozapine and we have included a statement in the methods section to describe this fact. We also mentioned it in the limitations section.

The psychiatric diagnosis of all the patients with TD was schizophrenia and we have included a statement in the methods section to describe it.

The pathway to clozapine treatment for TRS was at least one of the medications which failed was a second generation antipsychotic (risperidone or olanzapine) and we have included a statement in the revised manuscript to describe this.

---

## [Decision Letter · Decision Letter 1]

12 Feb 2025

PONE-D-24-21600R1Assessment of Treatment Outcomes and Associated Factors among Patients Treated with Clozapine at Amanuel Mental Specialized Hospital, Addis Ababa, Ethiopia.PLOS ONE

Dear Dr. Haile,

Thank you for submitting your manuscript to PLOS ONE. After careful consideration, we feel that it has merit but does not fully meet PLOS ONE’s publication criteria as it currently stands. Therefore, we invite you to submit a revised version of the manuscript that addresses the points raised during the review process.

We look forward to receiving your revised manuscript.

Kind regards,

Rakesh Karmacharya, MD, PhD

Academic Editor

PLOS ONE

Journal Requirements:

Reviewers' comments:

Reviewer's Responses to Questions

**Comments to the Author**

1. If the authors have adequately addressed your comments raised in a previous round of review and you feel that this manuscript is now acceptable for publication, you may indicate that here to bypass the “Comments to the Author” section, enter your conflict of interest statement in the “Confidential to Editor” section, and submit your "Accept" recommendation.

Reviewer #1: All comments have been addressed

Reviewer #2: All comments have been addressed

2. Is the manuscript technically sound, and do the data support the conclusions?

Reviewer #1: Yes

Reviewer #2: Partly

3. Has the statistical analysis been performed appropriately and rigorously? 

Reviewer #1: Yes

Reviewer #2: No

4. Have the authors made all data underlying the findings in their manuscript fully available?

Reviewer #1: Yes

Reviewer #2: Yes

5. Is the manuscript presented in an intelligible fashion and written in standard English?

Reviewer #1: Yes

Reviewer #2: No

6. Review Comments to the Author

Reviewer #1: Thanks for adequately addressing all the comments. It is good to see that the appropriate changes are made in the manuscript.

Reviewer #2: The authors need to clarify why patients with more than several months of clozapine were admitted to the hospital in that one month.

What was the average dose of clozapine prescribed to the patients?

What was the nature of the side effects reported?

Was agranulocytosis reported in any of the patients?

What was the adherence to monitoring of blood counts in the patients?

What psychoactive substances were the patients using?

The manuscript needs a thorough proof reading for English and typographical errors (Risperidong, Clozapie etc. amongst many).

7. PLOS authors have the option to publish the peer review history of their article (what does this mean? ). If published, this will include your full peer review and any attached files.

**Do you want your identity to be public for this peer review?** For information about this choice, including consent withdrawal, please see our Privacy Policy .

Reviewer #1: No

Reviewer #2: No

---

## [Author Response · Author response to Decision Letter 2]

6 Mar 2025

Dear reviewers, thank you for your constructive comments. I believe the comments have significantly improved the quality of the manuscript so far and will do so this time too. I have addressed the remaining comments as follows:

1.The reason why patients stayed in hospital for one month was clarified in the manuscript. The reason for one month admission was for initiation of clozapine. Once stabilized, the patients were discharged from hospital and continued their treatment as outpatient.

2.The average dose of clozapine was 296 mg and this was included in the results section, clinical characteristics subsection of the manuscript.

3.The nature of side effects identified was mild forms of sedation, hypersalivation and occasional urinary incontinence. A statement is added to results section, clinical characteristics subsection of manuscript.

4.No agranulocytosis was reported and all participants were continuing their treatment with monitoring. A statement is added to the results section, clinical characteristics subsection of manuscript.

5.There was no problem reported about compliance to blood count and ESR monitoring of the patients and this is addressed also in the manuscript.

6.The psychoactive substances used are already in the manuscript. The reported ones were khat and cigarettes.

7.We have gone through the manuscript for proof reading and we have made several editions to grammar and punctuation to the manuscript.

Finally, we hope this will satisfy the requirements based on your comments.

With regards

---

## [Editor Report · Decision Letter 2]

10 Mar 2025

Assessment of Treatment Outcomes and Associated Factors among Patients Treated with Clozapine at Amanuel Mental Specialized Hospital, Addis Ababa, Ethiopia.

PONE-D-24-21600R2

Dear Dr. Haile,

We’re pleased to inform you that your manuscript has been judged scientifically suitable for publication and will be formally accepted for publication once it meets all outstanding technical requirements.

Kind regards,

Rakesh Karmacharya, MD, PhD

Academic Editor

PLOS ONE
---

## [Editor Report · Acceptance letter]

PONE-D-24-21600R2

PLOS ONE

Dear Dr. Haile,

I'm pleased to inform you that your manuscript has been deemed suitable for publication in PLOS ONE. Congratulations! Your manuscript is now being handed over to our production team.

Kind regards,

on behalf of

Professor Rakesh Karmacharya

Academic Editor

PLOS ONE